# A Photonic Immunosensor Detection Method for Viable and Non-Viable *E. coli* in Water Samples

**DOI:** 10.3390/microorganisms12071328

**Published:** 2024-06-29

**Authors:** Ana Fernández Blanco, Yolanda Moreno, Jorge García-Hernández, Manuel Hernández

**Affiliations:** 1Lumensia Sensors S.L., 46020 Valencia, Spain; afernandez@lumensia.com; 2Institute of Water and Environmental Engineering, Universitat Politècnica de València, 46022 Valencia, Spain; 3Advanced Center for Food Microbiology, Biotechnology Department, Universitat Politècnica de València, 46022 Valencia, Spain; jorgarhe@btc.upv.es (J.G.-H.); mhernand@btc.upv.es (M.H.)

**Keywords:** *E. coli*, water pollution, food safety, immunosensor, detection method, bioreceptors

## Abstract

Detection and enumeration of coliform bacteria using traditional methods and current molecular techniques against *E. coli* usually involve long processes with less sensitivity and specificity to distinguish between viable and non-viable bacteria for microbiological water analysis. This approach involves developing and validating an immunosensor comprising ring resonators functionalized with specific antibodies surrounded by a network of microchannels as an alternative method for detecting and indirectly enumerating *Escherichia coli* in samples of water for consumption. Different ELISA assays were conducted to characterize monoclonal and polyclonal antibodies selected as detection probes for specific B-galactosidase enzymes and membrane LPS antigens of *E. coli*. An immobilization control study was performed on silicon nitride surfaces used in the immunosensor, immobilized with the selected antibodies from the ELISA assays. The specificity of this method was confirmed by detecting as few as 10 CFU/mL of *E. coli* from viable and non-viable target bacteria after applying various disinfection methods to water samples intended for human consumption. The 100% detection rate and a 100 CFU/mL Limit of Quantification of the proposed method were validated through a comprehensive assessment of the immunosensor-coupled microfluidic system, involving at least 50 replicates with a concentration range of 10 to 10^6^ CFU/mL of the target bacteria and 50 real samples contaminated with and without disinfection treatment. The correlation coefficient of around one calculated for each calibration curve obtained from the results demonstrated sensitive and rapid detection capabilities suitable for application in water resources intended for human consumption within the food industry. The biosensor was shown to provide results in less than 4 h, allowing for rapid identification of microbial contamination crucial for ensuring water monitoring related to food safety or environmental diagnosis and allowing for timely interventions to mitigate contamination risks. Indeed, the achieved setup facilitates the in situ execution of laboratory processes, allowing for the detection of both viable and non-viable bacteria, and it implies future developments of simultaneous detection of pathogens in the same contaminated sample.

## 1. Introduction

Water is commonly contaminated by enteric bacteria, which inhabit the human gastrointestinal tract and are excreted in fecal matter [1]. Consuming water contaminated with these microorganisms can lead to intestinal infections [2]. Total coliforms are a key indicator group for assessing water quality [3].

Traditional methods for detecting and enumerating coliform bacteria typically involve lengthy processes, including enrichment with culture media in the laboratory [4]. However, advancements in molecular techniques have introduced faster, more sensitive, and more specific methods for microbiological water analysis [5], many of which can be automated. These new techniques enable the identification of pathogens, including novel and emerging strains [6]. Consequently, portable biosensors are now being discussed for their high sensitivity in detecting low coliform bacteria concentrations in water samples [2].

Conventional enumeration of viable pathogens in the water relies on fecal indicator bacteria culture methods using selective media. Research has shown that bacteria, including pathogens like *Escherichia coli*, can enter a viable but non-culturable (VBNC) state under environmental stress and starvation conditions [7,8,9]. This state affects the accuracy of traditional enumeration methods for viable bacteria in water supplies and wastewater treatment systems. RT-PCR also quantifies the target organism’s DNA in real time, providing information on the concentration of pathogens in the sample. However, these methods can overestimate viable bacteria counts due to the persistence of DNA after cell death [10,11]. To address this, mRNA detection via RT-PCR has been developed to test the viability of VBNC bacteria [12,13].

Water quality testing often focuses on detecting *Escherichia coli*, considered the best indicator of fecal contamination [14,15,16,17,18]. According to WHO guidelines, water poses an intermediate risk when *E. coli* counts range from 10 to 100 CFU per mL and a high risk when counts are between 100 and 1000 CFU per mL [14].

Preferred methods for microbiological water quality testing include membrane filtration and colony counts. *E. coli* confirmation typically employs immunoassays (ELISA) and PCR with species-specific DNA primers [17]. Traditional culture-based methods allow for sufficient bacterial growth for detection and enumeration for at least 24 h [19]. These methods involve culturing the bacteria on selective media, followed by colony counting and biochemical testing to determine the concentration of viable cells in the sample [20]. Although practical and low-cost, these methods are time-consuming, less sensitive, and less specific.

Slow and time-consuming traditional methods can delay disease diagnosis and treatment [20]. They are also expensive and require highly trained personnel. Nucleic-acid-based strategies, particularly PCR, offer high sensitivity and rapid detection but are labor-intensive, time-consuming, and costly [21,22]. Despite these drawbacks, nucleic-acid-based methods are powerful tools for pathogen detection in food, overcoming long enrichment stages due to their ease of use and automation potential.

The current molecular methods based on microarrays, nucleic acid sequence amplification, PCR, PCR hybridization, and real-time PCR allow for the detection of various pathogens in the same test. These techniques enable high specificity and sensitivity rates, ensuring rapid and accurate identification of multiple pathogens [23]. In addition, using PCR for pathogen detection requires knowledge of target DNA sequences to design specific primers [24], which limits the technique’s detection capacity.

Multiplex PCR enables the detection of various pathogens, including *E. coli* strains and their verotoxins. Additionally, this technique can offer high sensitivity and specificity, making it a valuable tool for food safety testing [25]. However, inhibitors in samples can affect the technique’s sensitivity and specificity, leading to false negatives or positives. Additionally, multiplex PCR requires careful validation for each pathogen and toxin.

ELISA is a widely used technique in immunology and diagnostics due to its ability to detect and quantify specific *E. coli* antigens or antibodies in biological samples. ELISA typically takes about 24 h from sample preparation to results [26,27], though this can vary depending on sample preparation, incubation times, and detection methods [28,29]. ELISA quantitatively determines antigen concentrations in samples, with signal intensity from labeled antibodies correlating with the antigen amount. This quantitative aspect makes ELISA powerful in clinical diagnostics, immunology research, and biotechnology [30,31]. The method involves immobilizing the target antigen onto a solid substrate, typically a plastic surface, followed by detection using a labeled selective antibody [32]. The sensitivity of ELISA can be influenced by various factors, including the quality of antibodies used, the efficiency of antigen capture and detection, and the presence of interfering substances in the sample matrix. Therefore, validating the ELISA method thoroughly and comparing its performance against reference methods is essential to ensure reliable results.

Capture ELISA (cELISA) is a widely used technique for detecting *E. coli*. Through this method, antibodies that specifically target *E. coli* are fixed to a plate. When a sample containing *E. coli* antigens is introduced, these antibodies capture and bind the antigens, allowing for their detection. After capturing the antigens, primary antibodies specific to *E. coli* are added, followed by secondary antibodies. These secondary antibodies are linked to an enzyme, such as horseradish peroxidase (HRP), or directly conjugated to the detection molecule. Although cELISA offers rapid detection, it may sacrifice some sensitivity compared to more time-consuming methods, balancing speed and accuracy for various applications [33].

Direct ELISA (iELISA) simplifies assays by requiring only one antibody—the primary antibody binding to the target antigen—compared to cELISA, which involves two antibodies [30]. This approach streamlines the experimental setup and is suitable when a specific antibody is available for the antigen of interest.

Biosensors provide advantages over conventional detection methods, including speed and sensitivity [34,35]. They offer high specificity, detecting specific pathogens in water and food samples with precision [36]. Biosensors offer multiplexing. Additionally, biosensors can target various pathogens or contaminants simultaneously, reducing the timing of analysis and providing detection accuracy by providing comprehensive information about the sample’s microbial or contaminant profile [37], making them valuable tools in food safety and environmental monitoring.

Optical biosensors provide rapid and efficient responses for ensuring food safety, particularly in scenarios where timely detection is crucial to prevent outbreaks or minimize economic losses [38]. They can be integrated into automated systems, enabling real-time monitoring of low levels of target analytes and providing high sensitivity and specificity in food samples. In addition, optical biosensors offer portability and ease of use, making them well-suited for deployment in field environments [38].

Integrating photonic biosensors onto silicon integrated circuits or PICs revolutionizes point-of-care diagnostics by offering compact, sensitive, and multiplexed detection platforms highly compatible with CMOS technology [39]. Sensitivity is crucial for biosensor performance, as indicated by limits of detection (LoD) and quantification (LOQ). Enhanced sensitivity improves biosensor efficacy in detecting and measuring specific substances [40]. The sensor’s sensitivity is determined by how effectively the evanescent field interacts with the sample molecules [41,42].

Microfluidic-based biosensors with microchannels for fluidic samples are becoming increasingly relevant due to their on-chip immunoassay capabilities [43]. These systems perform various laboratory processes concurrently on a single chip, including detection, sampling, separation, and mixing [44,45]. Besides being extremely sensitive [46], biosensors can be incorporated into portable sensing systems [47]. Portable biosensors can determine spatio-temporal variations in water quality by deploying sensors in water sources or installing them at the point of source [47].

Ultra-sensitive biosensors with low LoD can detect single microbial cells. While this sensitivity is ideal for assessing water quality, the uneven spread of microbial cells in large volumes of water (e.g., lakes, rivers, wells) can affect accurate cell count. This criterion must be considered when designing biosensors for water quality assessment.

While optical microfluidic biosensors demonstrate competitive sensitivities, they still face challenges that must be addressed, including high costs and complex procedures for integrating optical components with microfluidic channels and functionalized surfaces [48]. Current *E. coli* biosensors still involve labor-intensive manual procedures, which can limit their practical use in laboratory settings.

This research aims to optimize a photonic immunosensor capable of detecting both viable and non-viable forms of *E. coli* in water samples for human consumption. The detection method uses ring resonator transduction within functionalized Photonic Integrated Circuits (PICs) coated with specific antibodies targeting lipopolysaccharides (LPSs) from the outer membrane of *E. coli* or *E.-coli*-specific β-galactosidase, a peptide carrier protein with biological activity. By enabling early detection and quantification of viable and non-viable bacteria, this technology promises to predict pathogen contamination in water samples, addressing current limitations in pathogen detection systems within the agrifood industry and environmental sector.

## 2. Materials and Methods

### 2.1. Antibodies and Reagents

#### 2.1.1. Immobilization Process

The reagents that were used to carry out the functionalization process were the following: 1% CTES (carboxyethylsilanetriol, disodium salt 25% in MilliQ water, ABCR, Karlsruhe, Germany), 0.1 M MES [2-(N-Morpholino)-ethanesulfonic acid] (ThermoFisher, Waltham, MA, USA), EDC [1-Ethyl-3-(3-dimethylaminopropyl)-Carbodiimide] (Sigma, San Luis, MO, USA), and NHS [N-Hydroxysuccinimide] (Sigma) as the coupling reagent.

A polyclonal (chicken) anti-B-Galactosidase antibody specific to *E. coli* (AB3403-I, Sigma-Merck, San Luis, MO, USA) and monoclonal (mouse) anti-*E. coli* antibody against membrane lipopolysaccharides (LPSs) (ab35654, Thermo Fisher Scientific, Cambridge, UK) constitute the antibodies against the target object of this study. These antibodies were tested against *E. coli* target antigens. The negative control in order to validate the results of our experiment involved using the specific rabbit polyclonal anti-fish antibody obtained from Eurofins Inmunolab (Reinbek, Germany).

#### 2.1.2. Indirect ELISA Procedure

Primary polyclonal antibodies, such as rabbit anti-*E. coli*, were used (GTX13626, GeneTex, Irvine, CA, USA).

Rabbit anti-*E. coli* serotype O/K (PA1-7213, ThermoFisher, Waltham, MA, USA), rabbit anti-*E. coli* serotype O/K (ab31499, Abcam, Cambridge, UK), and polyclonal (chicken) anti-B-Galactosidase (AB3403-I, Sigma-Merck, San Luis, MO, USA) were used.

As primary monoclonal antibodies, mouse anti-*E. coli* LPS (ab35654, Abcam) and mouse anti-*E. coli* (6911,35-660, Prosci-Inc., Poway, CA, USA) were used.

The specific antigens were different concentrations of LPS from *Escherichia coli* eBioscience™ (Thermo Fisher Scientific, Waltham, MA, USA).

Indirect ELISA reagents included H_2_SO_4_ solution, 0.1 M hydrochloric acid (Scharlab, Barcelona, Spain), and commercial TMB substrate (Thermo Scientific, Waltham, MA, USA).

The secondary antibodies used comprise GARPO polyclonal anti-rabbit IgG peroxidase (Abcam, Cambridge, UK) and GAMPO polyclonal anti-mouse IgG peroxidase (Abcam, Cambridge, UK).

Samples were inoculated in a pH 9.6 0.05 M carbonate buffer. This buffer was chosen to maintain a stable pH environment for antigen–antibody interaction. Each sample was inoculated in duplicate to ensure reproducibility and reliability of the results.

The carbonate buffer was sourced from Merck (Darmstadt, Germany). The negative control was PBS (phosphate-buffered saline) solution. A conventional solution of lipopolysaccharides (LPSs) from *Escherichia coli* was used as the positive control. LPS is a component of the outer membrane of Gram-negative bacteria like *E. coli* and serves as a reliable positive control to verify the sensor’s functionality and sensitivity.

Seven serial dilutions were created from 1 mL of *E. coli* antigens from strains 101, 425, 418, and 4558 (Spanish Type Culture Collection, CECT, Valencia, Spain). These dilutions were prepared in 0.05 M carbonate buffer pH 9.6, resulting in 10 to 10^5^ CFU/mL concentrations.

A stock solution of LPS *E. coli* antigen from Escherichia coli eBioscience™ was prepared, and the diluent used was 0.05 M carbonate buffer at pH 9.6. Seven serial dilutions of LPS *E. coli* antigen from *Escherichia coli* eBioscience™ (Termo Fischer Scientific, Waltham, MA, USA) were prepared at concentrations from 0.06 ng/mL to 4 ng/mL in 0.05 M carbonate buffer pH 9.6.

Absorbance measurements were conducted at two wavelengths, 450 nm (OD450) and 650 nm (OD650), with the use of the Varioskan Flash multimode scanning microplate reader (Multilabel Victor 1420 Counter). The criteria for positive control designation (P/N 2.1) were based on the OD450 values being 2.1 times higher than the negative controls.

#### 2.1.3. Sensor Detection Assay

The validation process ensues by preparing a range of serial dilutions of overnight *E. coli* cultures (strains 101, 425, 418, 4558) and inoculating them into *E.-coli*-free water, with concentrations from 10 to 10^6^ CFU/mL.

The biosensor’s response was evaluated with replicates of naturally contaminated water samples from a meat processing facility in the Valencia region, stored at 4 °C, and artificially contaminated water samples treated with UV radiation and stored similarly. These samples were inoculated with *E. coli* strains with 10 to 10^6^ CFU/mL concentrations.

Signal transduction connected to 100 Photonic Integrated Circuits (PICs) made by Lumensia Sensors (Spain) that are resonance-related involves the conversion of optical signals generated by the biosensor into measurable data (Figure 1) [49]. The setup involved integrating 100 photonic biosensors into silicon PICs connected to a two-channel microfluidic system and a peristaltic pump.

### 2.2. Binding Capacity According to the i-ELISA Method

An indirect enzyme-linked immunosorbent assay (ELISA) was developed to assess the binding capacity of selected primary polyclonal and monoclonal antibodies. The protocol followed previous descriptions with modifications [50,51]. First, 100 µL of the corresponding *E. coli* LPS or B-galactosidase concentrations were loading into each well of a 96-well ELISA microplate. Appropriate replicates and controls were included to assess antibody specificity, such as negative water samples spiked with *E. coli* cultures. Wells were washed three times with PBS containing 0.05% Tween-20 (PBS-T) to remove any unbound antigen or contaminants, and then they were blocked by adding 100 µL of 1% bovine serum albumin (BSA) in PBS to each well to block any remaining uncoated surfaces and prevent non-specific binding of antibodies. The microplate was incubated at room temperature (25 °C) for one hour to allow for complete blocking of the plate surface.

After blocking, a PBS solution containing 1 ppm of the selected *E. coli* antibody was prepared to add 100 µL of the prepared antibody solution to each well of the microplate. Then, the microplate was incubated for one hour at 37 °C to allow the primary antibody to bind specifically to the immobilized antigen on the plate. Wells without immobilized antibodies served as controls.

After washing three times with PBST to remove any unbound primary antibody, 100 µL of the goat anti-rabbit–HRP conjugate was applied to the wells. Finally, the microplate was incubated for one hour at 37 °C to allow the secondary antibody to bind specifically to the primary antibody. Detection was started by adding 100 mL of the OPD substrate solution into the microplate: 4 mg of o-phenylenediamine (OPD) and 15 µL of hydrogen peroxide (H_2_O_2_) in 10 mL of citrate buffer (pH 4.5). The microplate was incubated at room temperature in the dark for a suitable amount of time (usually 10–30 min) until the color developed. Finally, using a Variskan Flash microplate reader, the reaction was halted by introducing 50 µL of 2 M sulfuric acid and monitoring the absorbance of each well at 450 and 650 nm.

The sample was considered positive if its absorbance value at 450 nm was 2.1 times higher than the absorbance value of the negative control. This ratio is referred to as the Positive-to-Negative ratio (P/N ratio), with a threshold of 2.1 indicating positivity.

### 2.3. Immunosensor Fabrication and Antibody Functionalization

An electron beam writing technique was used to fabricate Optical Photonic Integrated Circuits (PICs) in a class 10–100 clean room using a 100 nm layer of PMMA (polymethyl methacrylate) positive resist onto the wafer [49]. The process began with preparing a silicon wafer, depositing positive photoresist and defining circuit patterns using lithography. After developing the photoresist, chromium was deposited via evaporation. The remaining photoresist was removed, leaving metal patterns. The result is a Photonic Integrated Circuit (PIC) with well-defined waveguide structures, encapsulated by a silicon oxide layer to cover the circuitry.

One hundred photonic biosensors (PICs) were functionalized with anti-*E. coli* antibodies: polyclonal (chicken) anti-B-Galactosidase (AB3403-I, Sigma-Merck) and monoclonal (mouse) anti-*E. coli* LPS (ab35654, Thermo Fisher Scientific). By following the inmobilitzation method [49], PICs were integrated into a two-channel microfluidic cartridge.

### 2.4. Photonic Lab On-Chip Technology

A photodetector converted the optical signals into electrical signals, which were then processed by a data acquisition system [49]. A high-resolution optical spectrum analyzer was used to measure the shifts in the resonance wavelength. The system continuously monitored the output spectrum of the resonators to detect any shifts indicative of antigen binding. The approach, developed by Lumensia Sensors, combines software and hardware to translate optical signals into resonance measurements, recorded in picometers (pm).

The photon transduction principle utilizing silicon nitride ring resonators (RRs) forms the foundation of this detection system. These ring resonators analyze refractive indices for various applications. In this case, as *E. coli* antigens bind to the anti-*E. coli* antibodies held on the ring resonators, the resulting change in the refractive index shifts the resonant wavelength of the light circulating within the rings. This shift is detected and measured using a photodetector and analyzed to determine the presence and concentration of the target antigens.

Integrating microfluidic systems allows for handling tiny sample volumes, reducing the biological material required for analysis. The label-free detection capability of this biosensor facilitates real-time monitoring of biological interactions. This dynamic observation is critical for applications requiring immediate feedback, such as continuous environmental monitoring [39].

This detection system captures biosensor’s transduction signal during sample analysis using a setup that integrates three essential components. The PIC, designed by Lumensia Sensors, is at the heart of the detection system. It contains ring resonators that are functionalized with specific antibodies for the target analytes. The microfluidic system consists of two channels that are integrated with the PIC. These channels allow for the simultaneous analysis of two separate samples. The peristaltic pump is used to control the flow of the sample through the microfluidic channels. It ensures a precise and consistent flow rate, which is critical for accurate sensor readings [49].

Each channel contains four ring resonators, allowing for parallel analysis and increasing the sensor’s throughput and efficiency. The sensor’s eight ring resonators are divided into two channels [52]. This multi-resonator, multi-channel configuration significantly enhances detection sensitivity, enabling the system to detect concentrations as low as ng/mL [49].

### 2.5. Sensor Validation for Viable and Non-Viable E. coli Cells

The performance and reliability of the photonic biosensor for detecting *E. coli* in water samples were rigorously evaluated through multiple experimental tests. These tests involved using various commercial strains of *E. coli*, which were spiked into samples of water for consumption to simulate real-world contamination scenarios. A microfluidic system passed the prepared water samples over the functionalized photonic biosensors. Controlled flow rates were maintained using a peristaltic pump to ensure consistent interaction between the sample and the sensor surface. Multiple experimental runs were conducted to establish the optimal sensitivity and Limit of Detection (LoD) for the photonic biosensor. Sensitivity was evaluated based on the smallest detectable concentration of *E. coli* that produced a significant signal shift compared to the baseline. The LoD was determined by analyzing the minimum concentration at which the sensor could reliably distinguish between contaminated and uncontaminated samples.

The detection of LPS and B-galactosidase as indicators of *E. coli*’s presence was validated, showcasing the biosensor’s ability to identify the bacteria’s structural and functional components. This made it possible to thoroughly assess how well the method worked for identifying and reacting to different *E.-coli*-specific antigens, such as certain LPSs and the B-galactosidase enzyme, thereby validating its use for viable and non-viable samples.

The analysis included different samples of water for consumption inoculated with *E. coli* strains 101 CECT, 425 CECT, 418 CECT, and 4558 CECT, ranging between 10 and 10^6^ CFU/mL, as explained in Section 2.2, to assess the sensor’s detection efficiency. The sensor’s performance was also evaluated by inoculating water samples with different dilutions of specific *E. coli* LPS and B-galactosidase to evaluate the immunosensor’s detection signals to both antigens. Additionally, serial dilutions of naturally contaminated water samples by *E. coli* within the same range were used to evaluate the immunosensor’s detection specificity.

To further validate the performance of the photonic immunosensor, the next step involved systematically flowing the inoculated water samples through a setup detector optimized for Photonic Integrated Circuits (PICs) made by Lumensia Sensors [53]. A microfluidic sticky layer enabled the regulated flow of samples, which was an essential component of the device. This layer ensured that the samples were consistently directed over the sensor surface, optimizing antigen–antibody interactions. The peristaltic pump maintained a consistent flow rate, ensuring uniform exposure of the samples to the sensor surface. Thus, the interaction of *E. coli* antigens with the functionalized ring resonators with antibodies on the PIC was monitored in real time.

To ensure the robustness and accuracy of the photonic immunosensor, a specific flowing protocol was implemented. This protocol aimed to establish a clear baseline signal, facilitate antigen–antibody interactions, and ensure thorough cleaning between sample runs. MilliQ water was initially run through the system for 3 min, establishing a reference signal by ensuring the sensor surface was free from contaminants and setting a baseline for subsequent measurements. After that, the sample of water for consumption was mixed with the bacterial sample and allowed to flow for 15 min. This duration allowed sufficient time for the *E. coli* antigens to interact with the functionalized immunosensors (ring resonators) on the PIC, ensuring optimal antigen–antibody binding and signal generation. Finally, a cleaning buffer flowed through the system for 5 min, clearing any residual materials and contaminants from the system, ensuring that the sensor surface was ready for the next sample without any carryover effects.

The photonic immunosensor system’s setup for reading resonance information is essential. It guarantees precise interpretation of the optical signals produced when the target analytes and biosensor interact. Lumensia Sensors’ sophisticated hardware and algorithms enable these optical signals to be translated into accurate and quantifiable resonance values, usually represented in picometers (pm). The monitoring and quantification of binding events on the surface of the biosensor is made possible by this system, which in turn yields important data regarding the concentration and presence of target analytes. To understand the optical signals produced during the interaction between the biosensor and the target analytes, the resonance data reading setup is essential [53].

The sensogram is an essential tool for analyzing the performance of the photonic immunosensor [53]. It provides real-time data on the sensor’s response to *E. coli* antigens, demonstrating its rapid, sensitive, and specific pathogen detection capability. Visualizing the differential resonance shifts makes it possible to accurately quantify the concentration of viable and non-viable *E. coli* in water samples, enhancing pathogen detection and contamination monitoring in various applications.

On the baseline of the X, the unit of measurement (time in seconds) represents the progression of the test over time, highlighting key phases, such as baseline establishment, antigen detection, and system cleaning. The resonance shift’s magnitude correlates with the antigen’s concentration, enabling quantification of *E. coli* in log CFU/mL. At the same time, on a Y basis (frequency difference in pm), the changes in resonance values indicate the interaction between the target antigen and the antibodies on the ring resonators.

### 2.6. Sensitivity and Specificity Statistical Evaluation

Through a double-blind experiment in which negative water samples were purposefully tampered with using various strains of *E. coli*, the method’s sensitivity and specificity were evaluated [52,54]. An analysis was performed to determine the significance of the findings. For every concentration, several iterations of the biosensor’s detection and quantification were carried out using the same reagents and equipment and under the same settings. This strategy validates the method’s sensitivity and specificity for identifying and measuring *E. coli* by guaranteeing the method’s consistency and dependability across different repetitions and concentrations.

An ANOVA test was used in the statistical analysis to determine the significance of each variable. Moreover, chi-square tests were performed at a 95% significance level to evaluate differences in the frequency of positive samples. Systat edition 9 software (SPSS Inc., Chicago, IL, USA) was used for data analysis. This statistical method aids in clarifying the importance of various factors and how they affect the results of the experiment. To ensure robustness and reproducibility in the statistical analysis, statistically significant differences, as indicated by a one-way analysis of variance (ANOVA), were taken into consideration when *p*-values were less than or equal to 0.05.

## 3. Results and Discussion

### 3.1. Sensitivity Assays through i-ELISA

This research aims to characterize antibodies for potential use in developing a biosensor against *E. coli*. This characterization was achieved by designing an indirect ELISA protocol that quantitatively assessed the binding capacity of the selected *E.-coli*-specific polyclonal and monoclonal antibodies. The protocol provides critical data on their effectiveness in the photonic immunosensor, as indicated by the previous scientific literature [30,31].

The *E. coli* antigen–anti-*E. coli* antibody binding complexes formed by the i-ELISA assay yielded different absorbance data. These values served to characterize the binding capacity of each antibody against the different *E. coli* antigens used as well as the specificity against each antigen. Thus, higher absorbance data coincide with higher specificity values against each target antigen, which means choosing the most specific and sensitive antibodies to different concentrations of antigen (Figure 2).

The absorbance values for monoclonal and polyclonal antibodies were plotted against the same *E. coli* strains and their specific LPS antigens. These curves highlight antibodies’ types of sensitivity and binding efficiency, enabling the quantification of the bacterial concentration in samples based on absorbance values. The specificity and sensitivity results from the i-ELISA immunoassay show that the selected antibodies have a high affinity for *E. coli* antigens. The standard curves generated from known concentrations of *E. coli* and its specific LPS provide a basis for using absorbance readings to calculate the amount of bacteria present in unknown samples. Monoclonal antibodies strongly bind to specific *E. coli* strains and LPS, as indicated by the high absorbance values at increased concentrations. Polyclonal antibodies also demonstrated effective binding, with variations in absorbance reflecting differences in binding affinity compared to monoclonal antibodies. These findings confirm the reliability of using these antibodies in the photonic immunosensor for detecting *E. coli* in water samples. This evaluation of each antibody’s binding capacity, specificity, and sensitivity aligns with previous studies, enhancing the reliability of these findings [33,55,56].

The optimal affinity results highlight the effectiveness of the selected antibodies in the photonic immunosensor system. The polyclonal antibody’s ability to bind efficiently across a wide range of concentrations without saturation suggests its robustness for diverse sample conditions. Meanwhile, the high binding efficiency of the monoclonal antibody at both low and high concentrations underscores its versatility and sensitivity, making it an excellent choice for a reliable and sensitive immunosensor probe [57].

### 3.2. Sensor Validation for Drinking Water

The newly invented photonic immunosensor for *E. coli* was tested for detection sensitivity over a concentration range of 10 to 10^6^ CFU/100 mL. The primary objective was to ascertain the most effective enrichment method for detecting *E. coli* in various samples, including both pure cultures and inoculated drinking water samples; multiple dilutions of *E. coli* were prepared, and pure bacterial cultures and drinking water samples inoculated with *E. coli* were used in the validation assays. Indirect quantification of *E. coli* concentrations was performed using the immunosensor. The results of this immunosensor quantification were benchmarked against gold standard quantification methods for *E. coli* (CFU/mL) (Appendix A).

Appendix A quantified dilutions of the *E. coli* strain CECT 425 with dilutions of this strain spiked in samples of water for consumption free of *E. coli*. This list supposes positive immunodetection results obtained for all samples of water for consumption in a range of 10 to more than 10^6^ CFU/100 mL of *E. coli* CECT 425, using the developed immunosensor across different operators and measurement equipment but on identical samples.

At the same time, the second objective was to validate the immunosensor under development in samples of water for consumption naturally contaminated by *E. coli* to test its effectiveness in a natural environment. For this, and as Appendix A shows, several quantified samples of serial dilutions of actual samples of contaminated water for human consumption employed in the food industry were tested. In this case, positive detection results were obtained for all samples contaminated by *E. coli* spp. with concentrations greater than 10 CFU/100 mL and up to 10^8^ CFU/100 mL across different operators and measurement equipment but on identical samples.

The Limit of Detection (LoD) is a pivotal metric for assessing a biosensor’s efficacy, indicating the minimum concentration of the target analyte that can be reliably identified. The developed photonic immunosensor demonstrated remarkable sensitivity, particularly for detecting *E. coli* in drinking water samples. For samples with as few as 10 CFU/mL of *E. coli*, the biosensor demonstrated a 100% detection rate. The ability to detect very low concentrations of *E. coli* underscores the biosensor’s applicability in real-world scenarios.

The statistical analysis of the biosensor’s detection method for *E. coli* in drinking water samples yielded a *p*-value of 0.0026. This result indicates a highly significant difference between the detection capabilities of the biosensor and random chance, underscoring the reliability and effectiveness of the method.

The biosensor is a reliable and accurate substitute for identifying *E. coli* in water samples, as demonstrated by the observed 100% agreement. Sensitivity refers to the biosensor’s ability to identify true positive cases correctly. The biosensor effectively detected samples contaminated with *E. coli* antigens and correctly identified samples free of the microorganism. Specificity reflects the biosensor’s ability to identify samples that do not contain the target microorganism accurately. The biosensor distinguished between true negatives and positives, confirming its ability to identify samples without *E coli* correctly.

A positive predictive value or PPV of 100% means that every time the biosensor identifies a sample as positive for *E. coli*, it is indeed a true positive. This eliminates false positives, ensuring that any detection by the biosensor is accurate. A negative predictive value or NPV of 100% means that every time the biosensor identifies a sample as negative for *E. coli*, it is indeed a true negative. This eliminates false negatives, ensuring that any non-detection by the biosensor is accurate. The PPV and NPV being at 100% highlights the biosensor’s reliability and accuracy in identifying both positive and negative samples.

Both methods, detection in artificially spiked samples and naturally contaminated samples, demonstrated high effectiveness in detecting *E. coli* when the water samples were intentionally contaminated with known concentrations of *E. coli*. This controlled testing environment allowed for precise calibration and validation of the detection methods.

Results demonstrate that detecting *E. coli* in drinking water samples, whether artificially spiked or naturally contaminated, is effectively achieved using either of the two tested methods. The comparable specificity levels indicate that both methods accurately detect the presence of *E. coli* antigens in drinking water samples.

The method consistently detected *E. coli* antigens in 98.7% of contaminated samples, indicating a high level of reproducibility. This implies that when examining the same samples at various times and under standard conditions, there is a 98.7% probability of obtaining the same detection result. Data in Appendix A summarize the reproducibility results for *E. coli* detection methods across various drinking water samples. Strong proof of the high repeatability and reliability of both *E. coli* detection methods can be seen in Appendix A. Consistent results across multiple tests and sample types ensure the methods can be trusted for routine water quality monitoring.

It is clear that the biosensor is sensitive to different *E. coli* concentrations from the correlation between the observed bacterial concentration and the optical signal it generates. This relationship is crucial for validating the biosensor’s effectiveness in detecting different concentrations of the target. Figure 3 provides a detailed illustration of the *E. coli* experiment’s microring resonance notch shift in picometers (pm), showcasing the biosensor’s response to various dilution factors and concentrations. Samples with larger concentrations of *E. coli* exhibit more prominent optical signals, which are represented as resonance notch shifts. This aligns with the principles of biosensing, where higher concentrations of the target analyte induce more significant changes in the optical properties of the sensor. Conversely, more diluted samples generate weaker optical signals, demonstrating the sensor’s ability to detect lower concentrations of *E. coli* in water samples effectively with dilution factors as low as 10 CFU/mL.

Calibration curves are fundamental to the immunosensor method, providing the means to quantify *E. coli* concentrations in unknown samples accurately. By establishing a clear relationship between resonance shifts and known concentrations, these curves enable the reliable assessment of food and water safety, ensuring the acceptability of products for human consumption [58].

Visual confirmation of linearity, supported by the calculation of correlation coefficients, adds credibility to the method’s quantitative capabilities, instilling confidence in using the immunosensor method for accurate and reliable quantification of *E. coli* in water samples. The three curves in Figure 3 confirm a linear relationship between the X variable (known concentrations) and the Y variable (measured resonance). This linear relationship is crucial for quantitative analysis, ensuring that the measured response is directly proportional to the analyte concentration.

To ensure the accuracy and reliability of the biosensor method for detecting *E. coli* in water samples, the linearity of the calibration curves was assessed in Figure 3. A visual inspection of these plots indicated a linear relationship between the concentrations and the resonance values across the tested range. The correlation coefficient (r) was calculated for each calibration curve to confirm linearity quantitatively. The regression equations and their respective r-values confirmed the linear relationship between the known concentrations and the measured resonance values. The working interval for each was established by analyzing the calibration curves, ensuring the method’s robustness across these ranges (10 to 10^6^ CFU/mL).

The LoQ (Lower Limit of Quantification) represents the lowest concentration of *E. coli* that can be reliably quantified with acceptable precision and accuracy using the biosensor. This threshold indicates that concentrations above 100 CFU/mL can be quantified with confidence, ensuring precise and accurate measurements. The LoD (Lower Limit of Detection) is the lowest concentration of *E. coli* that the biosensor can detect, although not necessarily quantified accurately. The method’s sensitivity for detecting the bacteria at concentrations as low as 10 CFU/mL demonstrates the biosensor’s ability to detect very low levels of *E. coli* in water samples, which is critical for early detection in contamination scenarios. The ULOQ (Upper Limit of Quantification) confirms the method’s robustness at higher concentration levels, maintaining accuracy and precision without saturation effects. This indicates that the biosensor can accurately quantify *E. coli* concentrations up to 10^6^ CFU/mL, ensuring a wide dynamic range for detection and quantification.

Results demonstrate that the immunosensor method is as effective as quantitative PCR (qPCR) and Reverse Transcriptase–PCR (RT-PCR) in detecting *E. coli* in drinking water samples. This suggests that the immunosensor achieves comparable accuracy and reliability to established molecular methods [59,60]. The immunosensor technology streamlines the analytical procedure in contrast to PCR-based techniques, which call for intricate bacterial processing stages, including buffers for lysis and purification of DNA kits. By directly detecting *E. coli* antigens using specific antibodies, the method eliminates the need for time-consuming and resource-intensive sample preparation steps. The use of specific antibodies in the immunosensor method enhances specificity, reducing the likelihood of false positives or nonspecific detections. Unlike PCR methods, which may occasionally yield false positives due to nonspecific amplification, the immunosensor’s antibody-based approach offers greater confidence in the results [61,62].

This immunosensor in development presents advantages related to the Limit of Detection (LoD), specificity, sensitivity, speed in testing, cost, and working interval, which make it a promising alternative to specifically detect *E. coli* if we compare it with other techniques that have already been developed (Table 1).

The label-free detection capability of the immunosensor method represents a significant advantage over labeled detection methods [63]. By eliminating the need for additional labeling or modification of the target analyte, the assay is simplified, and potential sources of variability are reduced [64]. The advantage of this immunosensor in development concerning labeling is reflected, for example, in the study led by Gutiérrez-del-Río in 2018 [65]. The underlying mechanism for the proposed *E. coli* measurement is the passive diffusion process of MUG and GUD suspended in gelatin, forming the blue fluorescence product 4MU in the channels, giving us positive reading results through a smartphone and ultraviolet light [66]. However, the chimeric proteins (ba GFP-hadrurin and GFP-pb5) failed concerning specificity and/or sensitivity; the chimeric protein (GFP-colS4) could perform a specific detection of *E. coli* in drinking water samples. In a procedure that lasted about 8 min for the final result, this biosensor protein could linearly detect between 20 and 103 CFU of these bacteria. Below 20 CFU, the system cannot differentiate the presence or absence of the target bacteria [65]. However, the qPCR model had an accuracy of 92% and 96% with the thresholds of 110 and 1000 cell equivalents (EC)/100 mL, respectively. The culture model had 90% accuracy in management decisions with the threshold of 110 MPN/100 mL [67].

**Table 1 microorganisms-12-01328-t001:** The results of the selected articles show that the studies have evaluated the methods for detecting *E. coli* in water samples.

Authors	Article Title	Outstanding Result	Year	Reference
Wandermur G.L., Rodrigues D.M.C., Queiroz V.M., Gonçalves M.N., Miguel M.A.L., Werneck M.M., Allil R.C.S.B.	Development of an immunosensor of plastic optical fiber for detection of microorganisms in water and environmental monitoring	The system is capable of detecting small changes in the refractive index in the external medium, varying the light intensity of the biosensor upon contact with suspensions of *Escherichia coli* at different concentrations.	2013	[68]
Nagalambika C., Murthy S.M.	Revalidation of testing methods for assessing microbial safety of groundwater	The study recommended that where there are laboratory facilities, MFT and MTFT testing should be carried out; however, in fields and in villages, H_2_ is fast and cheap. The S test should be used for the detection of fecal contamination in drinking water in locations where time, personnel, and laboratory facilities are very poor.	2013	[69]
Kim T., Han J.-I.	Fast detection and quantification of *Escherichia coli* using the base principle of the microbial fuel cell	The DTs of the laboratory samples were 140 min and 560 min for initial concentrations of 1.9 × 10^7^ CFU/mL and 42 CFU/mL at 44.5 °C. Furthermore, DTs for GUS assays were further shortened through induction with methyl β-D-glucuronide sodium salt (MetGlu).	2013	[70]
Stauber C., Miller C., Cantrell B., Kroell K.	Evaluation of the compartment bag test for the detection of *Escherichia coli* in water in 3M™ Molecular	The sensitivity and specificity were 94.9% and 96.6%, respectively.	2014	[71]
Gomi R., Matsuda T., Matsui Y., Yoneda M.	Fecal source tracking in water by next-generation sequencing technologies using host-specific *Escherichia coli* genetic markers	The combination of multiplex PCR and dual-index sequencing is effective in detecting multiple genetic markers in multiple isolates at the same time.	2014	[62]
Shaibani P.M., Jiang K., Haghighat G., Hassanpourfard M., Etayash H., Naicker S., Thundat T.	The detection of *Escherichia coli* (*E. coli*) with the pH sensitive hydrogel nanofiber-light addressable potentiometric sensor (NF-LAPS).	A supernernstian response of a 74 mV/pH change in NF-LAPS provides high sensitivity towards *E. coli* with a theoretical Limit of Detection (LOD) of 20 CFU/ml.	2016	[72]
Eltzov E., Marks R.S.	Miniaturized Flow Stacked Immunoassay for Detecting *Escherichia coli* in a Single Step	The analyte/antibody–HRP complex will generate a signal in contact with *Escherichia coli*; after optimization, the sensitivity of the immunoassay was adjusted to 100 cells mL^−1^. Primers showed specificities only for their corresponding target organisms.	2016	[63]
Gunda N.S.K., Dasgupta S., Mitra S.K.	DipTest: A litmus test for *E. coli* detection in water	It has been observed that different interfering contaminants have no impact on the DipTest, and it may become a potential solution to detect *E. coli* contamination at the point of origin.	2017	[73]
Gutiérrez-del-Río I., Marín L., Fernández J., Millán M.Á.S., Ferrero F.J., Valledor M., Campo J.C., Cobián N., Méndez I., Lombó F.	Development of a biosensor protein bullet as a fluorescent method for fast detection of *Escherichia coli* in drinking water	Two of the chimeric proteins (ba GFP-hadrurin and GFP-pb5) failed with respect to specificity and/or sensitivity, but the chimeric protein (GFP-colS4) was able to perform specific detection of *E. coli* in drinkable water samples in a procedure that took about 8 min for the final result. This biosensor protein was able to linearly detect between 20 and 103 CFU of these bacteria. Below 20 CFU, the system cannot differentiate the presence or absence of the target bacteria.	2018	[65]
Ozeh U.O., Nnanna A.G.A., Ndukaife J.C	Coupling immunofluorescence and optoelectrokinetic technique for *Escherichia coli* detection and quantification in water	This method has the potential to sensitively isolate *E. coli* from a large number of organic and inorganic contaminants in water in less than 4 h.	2018	[64]
Lacey R.F., Ye D., Ruffing A.M.	Engineering and characterization of copper and gold sensors in *Escherichia coli* and *Synechococcus* sp. PCC 7002	The fluorescence response of cyanobacterial sensors to gold was significantly reduced compared to that of analogous *E. coli*.	2019	[74]
Han E.J.Y., Palanisamy K., Hinks J., Wuertz S.	Parameter selection for a microvolume electrochemical *Escherichia coli* detector for pairing with a concentration device	The achievable detection time for a 1 CFU mL^−1^ simulated sample was 4.3 ± 0.6 h assuming no loss of performance in the filtration step.	2019	[75]
Bigham T., Dooley J.S.G., Ternan N.G., Snelling W.J., Héctor Castelán M.C., Davis J.	Assessing microbial water quality: Electroanalytical approaches to the detection of coliforms	Electrochemical techniques have numerous advantages over portable detection and, although a large number of approaches has been investigated, the use of galactosidase and glucuronidase assays predominates.	2019	[76]
Zarrinkhat F., Jofre-Roca L., Jofre M., Rius J.M., Romeu J.	Experimental Verification of Dielectric Models with a Capacitive Wheatstone Bridge Biosensor for Living Cells: *E. coli*	The theoretical model was validated by measuring changes in dielectric permittivity in a cell culture of *Escherichia coli* ATTC 8739 from WDCM 00012 Vitroids. The spheroidal model was confirmed to be more accurate.	2020	[77]
Rishi M., Amreen K., Mohan J.M., Javed A., Dubey S.K., Goel S.	Rapid, sensitive and specific electrochemical detection of *E. coli* using graphitized mesoporous carbon-modified electrodes	The GCE/GMC electrode showed excellent sensitivity and selective response towards *E. coli* 3 CFU/mL at 25.2 × 10^4^ CFU/mL and 252 CFU/mL at 2268 CFU/mL, respectively, with a Limit of Detection (LOD) of 50.40 CFU/mL.	2022	[78]
Treebupachatsakul T., Lochotinunt C., Teechot T., Pensupa N., Pechprasarn S.	Gelatin-Based Microfluidic Channel for Quantitative *E. coli* Detection Using Blue Fluorescence of 4-Methyl-Umbelliferone Product and a Smartphone Camera	The underlying mechanism for the proposed *E. coli* measurement is the passive diffusion process of the MUG secreted by *E. coli* and the GUD suspended in the gelatin, forming the blue fluorescence product 4MU in the channels, giving positive reading results from a smartphone and ultraviolet light.	2022	[66]

Conventional methods, such as concentrators, are comparable to the membrane filtration method to analyze the microbiological quality of stream water and water collected from the roof [71,79]. However, it takes a long time to obtain results. Therefore, molecular methods and biosensors like this are viable alternatives for field analysis of water-quality-indicator bacteria [70,80].

Compared to an optical immunosensor with a detection time and detection mechanism similar to the one developed, there is an immunosensor of plastic optical fibre [68]. However, for the developed system to be more applicable, there is a need to increase the detection limit of the biosensor.

The electrochemical test stands out among the novel techniques developed [76]. However, the test time of the method is even higher concerning this photonic immunosensor [75], which seems to be a solution to the sensitivity of the detection of *E. coli*. Even so, the sensitivity of the immunosensor studied here is greater [78].

Multiplex ring biosensors enable the simultaneous detection of various *E. coli* antigens, including both viable and non-viable forms [81,82]. By detecting multiple antigens concurrently, these biosensors streamline the detection process, eliminating the need for separate analyses. This efficiency improvement translates to faster results and reduced resource consumption, making microbial monitoring more accessible and cost-effective. Unlike traditional methods like PCR and ELISA, which necessitate separate analyses for each *E. coli* antigen, multiplex ring biosensors consolidate detection into a single assay. This consolidation simplifies workflow, minimizing complexity and accelerating sample processing [83,84].

The microfluidic device allows for parallel measurement of two samples by dividing eight annular resonators, each with four resonant rings, over two channels. This parallel processing capability significantly reduces the time required for analysis, enhancing throughput and efficiency. The system’s sensitivity to the ng/mL scale ensures accurate detection of target analytes even at low concentrations. With less than 30 min response times, the coupled microfluidic system delivers rapid results, enabling timely decision making.

With the ability to deliver results within 4 h, the biosensor enables swift identification of microbial contamination in the environmental sector or, indeed, in food samples. This rapid diagnostic capacity is crucial for ensuring water monitoring related to food safety or environmental diagnosis and allows for timely interventions to mitigate contamination risks.

## 4. Conclusions

This study highlights the possibility of a new photonic biosensor as a useful instrument for identifying *E. coli* in samples of water for consumption. The biosensor demonstrates impressive sensitivity, specificity, and accuracy, highlighting its capability to reliably detect both viable and non-viable *E. coli* at low concentrations.

The biosensor fabrication process selects silicon nitride for its optical properties and compatibility with CMOS processes, enhancing the performance and reliability of the biosensor. The functionalization of the immunosensor involved tailoring specific antibodies for detecting *E. coli* antigens, ensuring high specificity and sensitivity in samples of water for consumption. This approach is made possible by employing surface chemistry techniques to immobilize the antibodies on the silicon nitride surface, enhancing the binding efficiency and stability of the biosensor.

The initial validation phase of the photonic biosensor using 100 fabricated PICs (Photonic Integrated Circuits) demonstrated promising performance, reliability, and feasibility in detecting *E. coli* in samples of water for consumption. A microfluidic adhesive layer was integrated with the PICs to regulate the flow of samples over the sensor surface. This setup facilitated the controlled flow of samples, enhancing the interaction between the analytes and the biosensor. The integration of microfluidic systems and antibody functionalization, along with the sensitive and reproducible detection capabilities, demonstrate the potential of this innovative photonic biosensing approach for real-world applications in food safety and water quality monitoring.

Combining monoclonal and polyclonal antibodies enhances the immunosensor’s versatility, effectively allowing it to detect *E. coli* across a broad range of concentrations. This dual-antibody approach ensures high sensitivity at low concentrations and broad detection capabilities at varying levels. The specific performance of the monoclonal antibody at low concentrations makes it particularly useful for applications requiring precise detection of low levels of *E. coli*. In contrast, the polyclonal antibody’s broader binding range is beneficial for general detection purposes.

These insights contribute to developing a robust detection system, ensuring that the immunosensor method can be tailored to specific needs, whether for sensitive detection in low-contamination scenarios or broader surveillance in varied-contamination conditions. Overall, the results showed the biosensor method’s effectiveness in accurately detecting *E. coli* in samples of water for consumption, highlighting its ability to operate even at low concentrations.

## Figures and Tables

**Figure 1 microorganisms-12-01328-f001:**
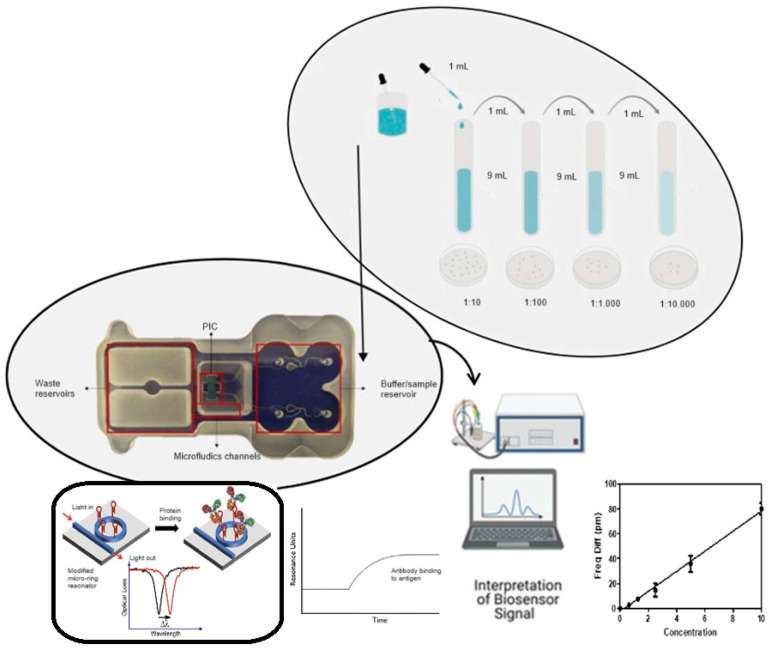
Scheme of the experimental process carried out in the validation of the immunosensor under development. (1) Preparation of serial dilutions of water samples contaminated artificially and naturally by *E. coli*. (2) Antigen–antibody immunoreaction on PIC. (3) Interpretation of the optical signal obtained after the binding of the *E. coli* antigen with the antibody against it deposited on the PIC, in concentration units.

**Figure 2 microorganisms-12-01328-f002:**
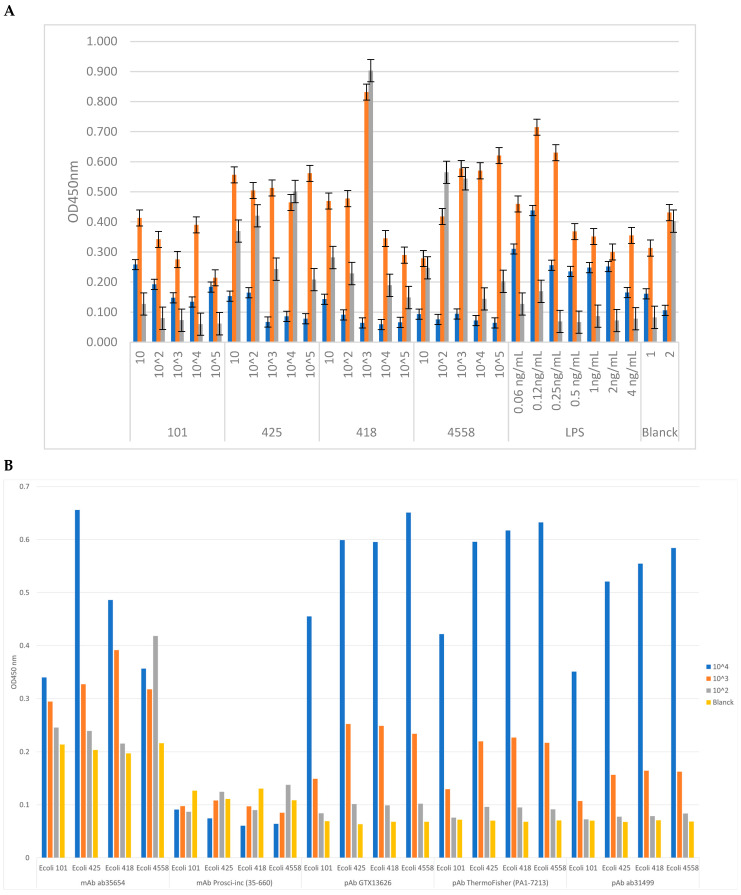
Evaluation of antibody *E. coli* response through i-ELISA immunoassay. (**A**) Absorbance results (OD 450 nm) for monoclonal anti-*E. coli* antibodies against different bacterial concentrations and dilution factors for commercial strains of *E. coli* (CECT 101, CECT 425, CECT 418, CECT 4558) and *E.-coli*-specific LPS. (**B**) Absorbance results (OD 450 nm) for both monoclonal and polyclonal antibodies against *E. coli* (CECT 101, CECT 425, CECT 418, CECT 4558) and *E.-coli*-specific LPS. This comparison illustrates both antibodies’ sensitivity and binding efficiency, providing insights into their effectiveness in the immunosensor.

**Figure 3 microorganisms-12-01328-f003:**
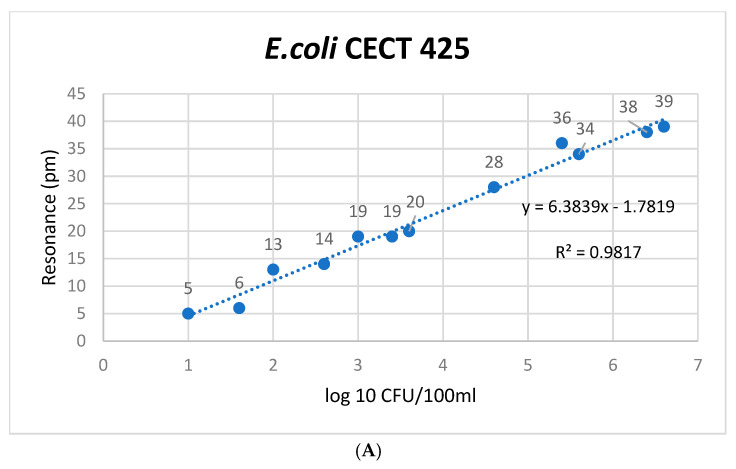
Validation calibration curves for *E. coli* antigens. These curves are divided into three panels. (**A**) Relationship between the resonance shift and the concentration of pure *E. coli* CECT 425 strain in a controlled environment. (**B**) Calibration curve of *E. coli* CECT 425 spiked in drinking water samples. (**C**) Calibration curve of naturally contaminated drinking water samples with *E. coli*. The Limit of Detection (LoD) and Limit of Quantification (LoQ) were computed to specify the minimum concentration of the analyte that the biosensor is capable of reliably detecting and quantifying. The Upper Limit of Quantification (ULOQ) is established in accordance with analysis requirements. In order to ascertain *s*0, no fewer than six determinations of samples at the calculated breakpoint concentration were performed. The concentration data for *E. coli* strains and both spiked and naturally contaminated samples were sourced from Appendix A, with resonance measured in picometers (pm) using a laboratory setup reader, as detailed in Section 2.

## Data Availability

Data are contained within the article.

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
