# Peer review of "A Photonic Immunosensor Detection Method for Viable and Non-Viable E. coli in Water Samples"

_microorganisms, 2024, doi:10.3390/microorganisms12071328_

Round 1

Reviewer 1 Report

Comments and Suggestions for Authors

The study conducted by Fernández et al. provides relevant information on the topic and can be considered for publication after some revisions.

The abstract needs to be rewritten. First, the authors should begin with a statement providing some background related to the topic and then clearly mention the main goals of the study. The main results (values) should be indicated and conclusions with practical implications as well as directions to future investigations should be provided.

The Introduction is adequate and provides the essentials for understanding the developed study.

In the Materials and Methods section, it would be helpful to provide a flowchart with all the steps taken in the investigation.

In the Results, please elaborate on some Tables, highlighting the obtained results.

Comments on the Quality of English Language

Minor editing of English language required

Author Response

Response to Reviewer 1 Round 1

Comments:

Point 1: The study conducted by Fernández et al. provides relevant information on the topic and can be considered for publication after some revisions. The abstract needs to be rewritten. First, the authors should begin with a statement providing some background related to the topic and then clearly mention the study's main goals. The main results (values) should be indicated, and conclusions with practical implications and directions for future investigations should be provided.

Response 1: Thanks for your contribution. In response to your appreciation, we have included, as you indicated in the established order, first the background, clear objectives of the study and then results with the data obtained and conclusions with practical implications, as well as possible developments of the technology in the immediate future.

Point 2: The Introduction is adequate and provides the essentials for understanding the developed study.

Point 3: In the Materials and Methods section, providing a flowchart with all the steps taken in the investigation would be helpful.

Response 3: Referring to his proposal, we have included a diagram that helps to understand the steps carried out in preparing the samples used and the validation developed for the method under study.

Point 4: In the Results section, please elaborate on some tables highlighting the results obtained.

Response 4: Thanks again for your appreciation. In response to your request, In the results section, we have added an explanation of some Tables, highlighting the results obtained.

Reviewer 2 Report

Comments and Suggestions for Authors

Interesting paper on detection of E. coli using a photonic biosensor.

Authors need to discuss their results better in instroduction and discussion.

Please use 106 cfu/ml throughout the text,

Please use italics for E. coli throughout the text.

Please use error bars for Figure 2.

What is the difference between LOD and LOQ? according to the definitions provided they refer to a minimum concentration of the analyte (E. coli).

Comments on the Quality of English Language

English is fine minor issues detected

Author Response

Response to Reviewer 2 Round 1

Comments:

Point 1: Interesting paper on detection of E. coli using a photonic biosensor. Authors need to discuss their results better in the introduction and discussion.

Response 1: Thanks for your contribution. In response to your request, we have added some paragraphs, including new bibliographical citations, that allow us better to explain the aspects of the introduction and the results.

Point 2: Please use 106 cfu/ml throughout the text.

Response 2: We have modified the value throughout the text in accordance with the reviewer’s recommendations.

Point 3: Please use italics for E. coli throughout the text.

Response 3: All the E. coli terms in the text are now in italics.

Point 4: Please use error bars for Figure 2.

Response 4: Thanks for your contribution. In response to your appreciation, we have included, as you indicated, the error bars in Figure 2.

Point 5: What is the difference between LOD and LOQ? The definitions provided refer to a minimum analyte concentration (E. coli).

Response 5: The essential difference between both parameters is that the Limit of Detection (LoD) is the one that allows the detection of the presence of the microorganism in the sample. The Limit of Quantification (LoQ) allows not only the detection of its presence but also the quantification of the quantity in which it is found in the problem sample.

Round 2

Reviewer 1 Report

Comments and Suggestions for Authors

I’m satisfied with the improvements made by the authors.

Reviewer 2 Report

Comments and Suggestions for Authors

Authors have revised and paper can be accepted

Comments on the Quality of English Language

Authors have revised and paper can be accepted